# Theoretical Analysis of a Magnetic Shielding System Combining Active and Passive Modes

**DOI:** 10.3390/nano14060538

**Published:** 2024-03-19

**Authors:** Qingzhi Meng, Zelin Wang, Qijing Lin, Dengfeng Ju, Xianfeng Liang, Dan Xian

**Affiliations:** 1State Key Laboratory for Manufacturing Systems Engineering, Xi’an Jiaotong University, Xi’an 710049, China; qzmeng2022@xjtu.edu.cn (Q.M.); wzl15086927209@stu.xjtu.edu.cn (Z.W.); danxian@xjtu.edu.cn (D.X.); 2School of Instrument Science and Technology, Xi’an Jiaotong University, Xi’an 710049, China; 3School of Mechanical and Manufacturing Engineering, Xiamen Institute of Technology, Xiamen 361021, China; 4Shandong Laboratory of Yantai Advanced Materials and Green Manufacturing, Yantai 265503, China; 5Xi’an Jiaotong University (Yantai) Research Institute for Intelligent Sensing Technology and System, Xi’an Jiaotong University, Xi’an 710049, China; 6Institute of Electric Power Sensing Technology, State Grid Smart Grid Research Institute Co., Ltd., Beijing 102209, China; judengfeng@geiri.sgcc.com.cn (D.J.); liang.xi@northeastern.edu (X.L.); 7State Grid Corporation Electric Power Intelligent Sensing Technology Laboratory, Beijing 102209, China

**Keywords:** electromagnetic shielding system, active and passive modes, magnetic shielding efficiency

## Abstract

Considering the magnetic shielding requirements of both geomagnetic field and 50 Hz power-line frequency in the complex working conditions of the power grid, an electromagnetic shielding system combining active and passive modes is proposed in this article. A three-dimensional Helmholtz coil with a magnetic shielding barrel nested inside is established by the COMSOL simulation tool, and the magnetic shielding efficiency of the system is analyzed. Comparing different materials, the simulation results indicate that permalloy alloy exhibits better shielding performance than pure iron and nickel materials. Additionally, the overall shielding efficiency of the shielding barrel increases linearly with the number of multiple layers. Under the combined active and passive electromagnetic shielding conditions, the system achieves a shielding efficiency of *S_E_* = 113.98 dB, demonstrating excellent performance in shielding both AC and DC interference magnetic fields. This study provides theoretical guidance for the construction of magnetic shielding systems in electromagnetic interference environment.

## 1. Introduction

With the rapid development of electronic devices and wireless communication technology, electromagnetic waves exist in various fields such as mobile phones, satellite communication, navigation, and medical equipment. These electromagnetic waves ensure the normal operation of electronic devices, but they also accumulate electromagnetic radiation in space. Unnecessary electromagnetic radiation can interfere with other electronic devices, which will reduce device sensitivity [1,2,3] and even lead to malfunctions of devices. In the power net, numerous electronic apparatuses like generators, substations, and cables are known to emit electromagnetic radiation, with 50 Hz power-line interference being particularly prevalent. Due to the make-before-break of high-power switches in the power grid, interference pulses frequently occur, resulting in the appearance of weak interference magnetic fields [4]. In addition, the geomagnetic field is always present, so the main sources of magnetic noise are geomagnetic and 50 Hz power-line magnetic field interference, which needs to be shielded.

At low frequencies, we can regard the magnetic field as a result of the electric current flow and the magnetization of surrounding materials. There are two basic methods for shielding against low-frequency magnetic sources: the diversion of the magnetic flux with high-permeability materials and the generation of opposing flux via Faraday’s law [5]. Therefore, we can combine the two methods to achieve active and passive shielding. Static and 50 Hz power-line magnetic fields belong to low-frequency magnetic fields, and high-permeability magnetic materials are usually used as shielding materials to ensure that the magnetic field lines gather inside the magnetic shielding material and do not leak into the external area, thereby achieving a significant attenuation of the magnetic field [6]. Traditional soft magnetic materials have good shielding effects. However, the shielding coefficient does not remain constant due to the nonlinearity of permeability and the closed loop characteristics of magnetic field lines [7,8], particularly when the external magnetic field intensity reaches excessive levels. In such cases, the shielding material may reach magnetic saturation and leads to a reduction in its effectiveness. Therefore, alloy materials are commonly used for shielding static and low-frequency magnetic fields. X. Ni et al. [9] used plasma spraying technology to prepare an amorphous Fe_73_._5_Cu_1_Nb_3_Si1_3_._5_B_9_ coating on a brass substrate to improve the shielding efficiency of low-frequency magnetic fields. At a thickness of 0.45 mm, the single-layer material has a shielding efficiency of *S_E_* = 10–12 dB. X. Huang et al. [10] prepared Fe–Ni foam materials for shielding in the lower frequency range and explored their shielding performance. The composite perforated foam material has excellent magnetic shielding performance in the frequency range of 10–500 kHz, and the single-layer material with a thickness of 0.25–0.36 mm has a shielding efficiency of *S_E_* = 20–60 dB. B.J. Madhu et al. [11] prepared graphene oxide by using graphite powder as the raw material and studied the electromagnetic shielding efficiency of graphene oxide/polyvinyl alcohol (PVA) nanocomposites, which exhibit a high dielectric constant below 100 Hz, with a maximum value of 38.87 dB at 50 Hz, which decreases as the frequency increases, indicating excellent low-frequency electromagnetic shielding effectiveness. In regard to numerical simulation, different shielding structures and materials have also been proposed. M. Bajtos et al. [12] proposed a Mu metal cage shielding structure and numerically investigated its shielding performance, obtaining a maximum shielding efficiency of 90%; J. Pang et al. [13] investigated carbon nanotubes (CNTs)/sodium alginate (SA)/ polydimethylsiloxane (PDMS) composite films containing Ni particles by simulation and calculation, and found that the shielding effectiveness can reach 50 dB by optimizing the cladding rate and size of Ni particles; L. Gozzelino et al. [14] examined a MgB_2_ bulk cylinder passive magnetic shielding material via simulation and achieved a shielding factor higher than 175 (equal to 44.9 dB shielding efficiency) at 20 K, and the results were reproduced by measurement. Although the shielding effect of single-layer composite materials is high enough, it is still difficult to achieve a magnetic shielding efficiency of more than 100 dB at low-frequency magnetic fields. However, by combining Helmholtz coils with magnetic shielding materials, the magnetic shielding efficiency can be greatly improved through actively compensating with the Helmholtz coils and passively shielding with the magnetic shielding materials.

In response to the magnetic shielding requirements of geomagnetic fields and 50 Hz power-line magnetic field interference under complex operating conditions in the power net, this paper proposes an electromagnetic shielding system containing a three-dimensional Helmholtz coil and a multi-layer metal magnetic shielding barrel, which combines the active and passive methods. The magnetic shielding efficiency of the system was analyzed utilizing the COMSOL simulation tool. A three-dimensional Helmholtz coil structure was established, and the distribution of the internal magnetic field was analyzed. The magnetic shielding efficiency of the AC and DC magnetic field of the system was simulated and compared with the current magnetic shielding system. This study provides theoretical guidance for the construction of magnetic shielding systems for magnetic–electric sensor modules.

## 2. Basic Theory of Magnetic Shielding

The shielding ability of materials can be represented by the electromagnetic shielding efficiency (*S_E_*), which is a function of the ratio of incident energy to residual energy, generally expressed in decibels (dB). *S_E_* is an important parameter for evaluating the magnetic shielding performance. It can be calculated using the following formula:$$S_{E}=10lg\frac{P_{0}}{P_{1}}=20lg\frac{H_{0}}{H_{1}}$$
where *P*_0_ is the power of the plane wave before shielding, *P*_1_ is the power of the plane wave after shielding, and *H*_0_ and *H*_1_ are the strength of the magnetic field before and after shielding, respectively.

Passive magnetic shielding through a shielding barrel is one of the most effective and fundamental methods for reducing magnetic fields. Figure 1a shows the principle of static and low-frequency magnetic shielding. When a shielding material is placed in a magnetic field, most of the magnetic field lines are transmitted along the shielding material due to its low magnetic resistance, rather than leaking into the external area, thereby achieving a significant attenuation of the magnetic field. The magnetic resistance of the shielding material is affected by its thickness and permeability, and the magnetoresistance *R_m_* can be expressed as follows:$$R_{m}=\frac{l}{\mu_{r}\mu_{0}S}$$
where *μ*_r_ is the relative permeability, *μ*_0_ is the permeability of the vacuum (equals to 4π × 10^−7^ N/A^2^), *S* is the area of the shielding material perpendicular to the direction of magnetic flux, and *l* is the average length of the magnetic field lines passing through the shielding materials.

Ferromagnetic materials with high permeability are usually used to shield low-frequency magnetic fields because of their low magnetic resistance. When an external magnetic field is applied to the shielding barrel, the magnetic field lines pass through the high permeability material and do not enter the shielded area, as shown in Figure 1a. The magnetic field line is a continuous closed curve, which is similar to an electric circuit obeying Kirchhoff’s law [15], as shown in Figure 1b, where *R*_m_ and *R*_0_ are the magnetic resistance of the shielding material and the free space, respectively. The shielding material with higher magnetic permeability shows better shielding efficiency. Furthermore, increasing the shielding material thickness can also reduce the magnetic resistance and increase shielding efficiency.

For active magnetic shielding, according to Biot-Savart’s law, the strength of a uniform magnetic field produced by a stable current on the axis of a coil is as follows:$$B=\frac{1}{2}\mu_{0}NIR^{2}\left\{{\left[R^{2}+{\left(\frac{R}{2}+x\right)}^{2}\right]}^{-\frac{3}{2}}+{\left[R^{2}+{\left(\frac{R}{2}-x\right)}^{2}\right]}^{-\frac{3}{2}}\right\}$$
where *R* is the average radius of the coil, *N* is the number of turns, *I* is the current passing through the coil, and *x* is the distance from a point on the axis to the center of the circle. To shield against 50 Hz power-line magnetic fields, the magnitude and direction of the three-dimensional coil current should be selected based on the amplitude and direction of the magnetic field to achieve active magnetic shielding. However, in some special environments, there may be interference from other frequencies or board band frequencies. In this case, there exists a dominant frequency at a broad band spectrum, and the current frequency of the coil should be adjusted to the dominant frequency for compensating, and other weaker interference frequencies can be eliminated through passive shielding.

## 3. Simulation of Magnetic Shielding Efficiency

### 3.1. Simulation of Passive Magnetic Shielding Efficiency

The passive magnetic shielding of magnetic sensitive components or modules is usually achieved through the use of a magnetic shielding barrel. A multi-layer magnetic shielding barrel structure was established by the COMSOL simulation tool, as shown in Figure 2. COMSOL uses finite element methods to solve electromagnetic fields in the simulation domain. Simulations used the three-dimensional MAXWELL equation as the basic model for simulation analysis and calculation, expressed as follows:$$\nabla \cdot E=\frac{\rho }{\varepsilon_{0}}\;\text{(Gauss’s law for electric field)}$$
∇·B=0 (Gauss’s law for magnetism)
$$\nabla \times E=-\frac{\partial B}{\partial t}\;\text{(Electromagnetic induction)}$$
$$\nabla \times B=\mu_{0}J+\mu_{0}\varepsilon_{0}\frac{\partial E}{\partial t}\;\text{(Ampere’s loop law)}$$where ***E*** is electric field intensity, ***H*** is magnetic field intensity, ***B*** is magnetic flux density, ***ρ*** is electric charge density, and ***J*** is current density. The following equations are used for calculating the magnetic flux density inside the shielding layer:$$n\left(B_{1}-B_{2}\right)=\nabla_{t}\left(d_{s}B_{t}\left(H_{t}\right)\right)$$
$$H_{t}=-\nabla_{t}V_{m}$$
$$B_{t}=\mu_{0}\mu_{r}H_{t}$$
where ***B*_1_** and ***B*_2_** are magnetic flux density of two nodes in the layer, $\nabla_{t}$ represents a tangential derivative (gradient), *d_s_* is the layer thickness, $\mu_{r}$ is the relative permeability, and *V_m_* is the magnetic scalar potential.

The electromagnetic shielding efficiencies of Fe, Ni, and permalloy alloy (Ni–Fe) materials were simulated by applying a DC geomagnetic field with an intensity of 5 × 10^−5^ T, as shown in Figure 3a. Due to the higher magnetic permeability of permalloy alloy compared to traditional Fe and Al pure metals, it can achieve higher shielding efficiency when shielding static magnetic fields. There is a gradient distribution of Fe and Ni elements inside the permalloy due to the mutual diffusion between Ni and Fe, and the Ni content in the Fe–Ni alloy layer gradually decreases from the surface to the interior, which can be regarded as a surface layer and a transition layer. Therefore, from a microscopic point of view, permalloy alloy can be regarded as a combination of multiple layered materials. Compared with single-layer materials, permalloy alloy exhibits enhanced shielding efficiency because of its higher magnetic permeability and the parallel connection of magnetoresistance in each layer, as shown in Figure 3b. The material source can be purchased commercially and the manufacturing processes of permalloy are undertaken via the deposition method [16]. Commercial electromagnetic pure iron (DT4E) in the form of cold-rolled plate was chosen as the substrate material. Ni layers were deposited on the two sides of pure Fe substrate to form a Ni/Fe/Ni diffusion couple. With temperature (45–60 °C) and cathodic current density (1–2.5 A/dm^2^), the permalloy is produced. Even though the cost of permalloy is higher than Cu, Fe, and Al materials, it shows a much better shielding performance. 

In our simulation, each thickness of the shielding layer is *d_s_ =* 3 mm, and the relative permeability for permalloy is $\mu_{r}$ = 20,000. The magnetic shielding efficiency of different numbers of permalloy magnetic shielding layers is shown in Figure 3c. It can be observed that with the increase in magnetic shielding layers, the magnetic shielding efficiency also maintains a linear increasing trend. However, with the increasing of layers of shielding material, the cost, volume, and the mass of the system will increase. So, the multiple layers cannot be added without limits. The shielding efficiency of multi-layer magnetic shielding can be expressed as follows:$$S_{E}=20lg\frac{H_{0}}{H_{s}}=20lg(1+\frac{nR_{0}}{R_{m}})$$

Increasing the number of permalloy layers can increase the deflection attenuation of magnetic field lines in the material, thereby improving the magnetic shielding efficiency. Besides, in the actual environment, the shielding materials and coils may experience wear and tear and performance degradation. The temperature variations, humidity, and mechanical stresses have a little influence on *R*_0_ and *R_m_*. The Curie temperature of the permalloy alloy reaches 410 °C. Under the Curie temperature, the materials exhibit ferromagnetism with high magnetic permeability, which means excellent shielding performance. High humidity can cause the corrosion of materials, leading to equipment degradation. Mechanical stress can cause system loosening or even fracture. Therefore, protective layers can be added to magnetic shielding materials and coils, and appropriate structural support and installation can be provided for the system to improve its reliability and avoid these problems. After long-term use, when the shielding performance of the system is found to be decreased, the coils or the shielding barrel needs to be replaced, and the old ones will be recycled. 

Figure 4 shows the distribution of the magnetic field perpendicular to the axis of the magnetic shielding barrel. It is observed that the magnetic field decreases gradually from the outside to the inside across the 6-layer permalloy alloy, which means that each layer of magnetic shielding material plays a shielding role, leading to a gradual weakening of the magnetic field.

The magnetic shielding efficiency may reach saturation in the presence of a strong external magnetic field, resulting in the failure of magnetic shielding. Therefore, the Helmholtz coil is carried out to compensate for the interference magnetic field, ensuring a higher magnetic shielding efficiency for the system.

### 3.2. Active Magnetic Shielding of Helmholtz Coil

The three-dimensional Helmholtz coil structure is shown in Figure 5. It is symmetrically nested in the x–y, x–z, and y–z planes, which have the same size of equivalent side lengths of 2*R* = 1500 mm, and the turn number of each internal coil is *N* = 100.

First, the uniformity of the magnetic field in each axis direction of the three-dimensional coil was simulated and analyzed. When a uniform current of 10 mA was applied to the six coils in the three-axis direction, the distribution of the magnetic field in each direction was measured and the results are shown in Figure 6. The three-dimensional Helmholtz coil produced a uniform magnetic field along the three-axis direction. In the three-axis direction, the magnetic flux density components are the same, producing a uniform magnetic field within the range of 250 mm × 250 mm × 250 mm, with a strength of *H* = 5 × 10^−5^ T. The magnetic field uniformities in the X, Y, and Z directions are 0.14%, 0.0614%, and 0.08%, respectively. 

When the Helmholtz coils are used for the compensation of the external magnetic field, the direction and magnitude of the magnetic field produced by the coil are opposite to the external magnetic field, thereby canceling the external magnetic field and achieving active magnetic shielding.

The active magnetic shielding of the geomagnetic field was achieved using the three-dimensional Helmholtz coil. Since the geomagnetic field was parallel to the z-axis, a current of 437 mA was applied to the z-axis coil to generate a magnetic field canceling the geomagnetic field. The magnetic field distribution after shielding is shown in Figure 7. Even though most of the geomagnetic field is shielded in the Helmholtz coil, there is still a relatively large residual magnetic field. This method can be only used to compensate for excessively strong external magnetic fields. To further improve the magnetic shielding efficiency of the system, the combination of active and passive methods needs to be adopted.

### 3.3. Simulation of Magnetic Shielding Efficiency for the Combined Active and Passive Mode

To establish a magnetic shielding system combining active and passive mode, a magnetic shielding barrel is placed inside a three-dimensional coil (as shown in Figure 8). The magnetic shielding efficiency of the system is simulated under a geomagnetic field and a 50 Hz power-line frequency interference magnetic field. Figure 9 shows the magnetic shielding results of the system against the geomagnetic magnetic field after overall compensation along the z-axis. The *S_E_* of the system reaches 87.86 dB, which is better than only using active shielding.

Besides the geomagnetic field, there is also interference from the alternating magnetic field generated by the 50 Hz power-line frequency current under actual working conditions. This system achieves better shielding efficiency against the alternating magnetic field through the combination of active and passive shielding by the three-dimensional Helmholtz coil together with the shielding barrel. When an external magnetic field with a frequency of 50 Hz and an amplitude of 5 × 10^−5^ T is applied, the transient analysis results for one period are shown in Figure 10. A maximum value of *S_E_* = 113.98 dB is achieved, which has a better shielding efficiency than the geomagnetic field. 

A comparison of the magnetic shielding efficiency between the combined active/passive magnetic shielding system and other types of magnetic shielding systems is presented in Table 1. The Ti_3_C_2_T_x_ MXene/Ni hybrid nanostructure forms numerous interfaces between Ni nanoparticles and MXene nanosheets, which greatly enhances polarization and exhibits excellent electromagnetic absorption and reflection capabilities. With a thickness of 1.8 mm, the shielding efficiency can reach 41.7 dB, which is a significant improvement compared to the single-layer polyimide material with a shielding effectiveness of 26.1 dB. The double-layer cylindrical shield material composed of superconducting material and iron achieves a shielding efficiency of 50 dB at 50 Hz, which is a significant improvement compared to the 18 dB of a single-layer superconducting material. At low frequencies, materials with high magnetic permeability exhibit better shielding efficiency. This is because single-layer superconducting materials are easily magnetically saturated in strong magnetic fields, while iron has high absorption loss and provides good shielding efficiency against low-frequency strong magnetic fields. Combining active compensation and passive shielding, the overall shielding efficiency is far exceeded than a passive magnetic shielding system. 

In general, the above simulation results show that the active and passive shielding system can achieve both geomagnetic field and 50 Hz frequency magnetic field shielding simultaneously. It provides theoretical guidance for the construction of magnetic shielding systems. Furthermore, the proposed magnetic shielding system can be miniaturized to adapt to different applications like medical devices, aerospace, and other related fields. It should be noted that in real applications, to avoid the potential electromagnetic radiation damage and disruption of the operation of surrounding equipment, a safe distance needs to be kept and operators should wear shielding clothing and keep related instruments away from the magnetic field during operation. By controlling the coil voltage and current within a safe range, the operation can maintain electrical safety and avoid fire hazards. 

## 4. Conclusions

A magnetic shielding system combining active and passive modes is proposed in this paper. The magnetic shielding efficiency of the system was analyzed by using COMSOL simulation tools. A three-dimensional coil structure was established and the uniformity of the magnetic field inside the coil was better than 0.16%. The active and passive magnetic shielding efficiency of the three-dimensional coil and the magnetic shielding barrel were simulated separately, as well as the magnetic shielding efficiency of the overall system. The shielding efficiency against DC magnetic fields reached 87.86 dB, with an improvement by up to 46%. The shielding effectiveness against 50 Hz AC magnetic fields reached over 100 dB. This work provides theoretical guidance for the magnetic shielding of magnetic sensor modules and power equipment. In addition, it can also be extended to fields such as medical devices or aerospace.

## Figures and Tables

**Figure 1 nanomaterials-14-00538-f001:**
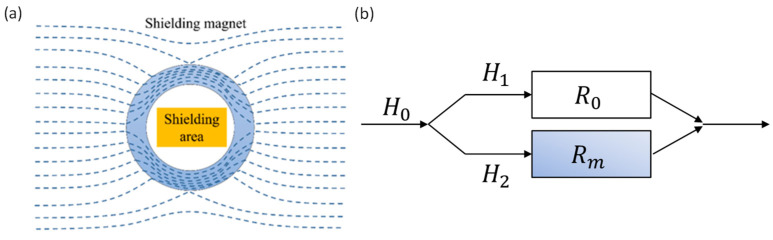
(**a**) Schematic diagram for the principle of passive magnetic shielding [15]. (**b**) Schematic diagram of magnetic field diversion.

**Figure 2 nanomaterials-14-00538-f002:**
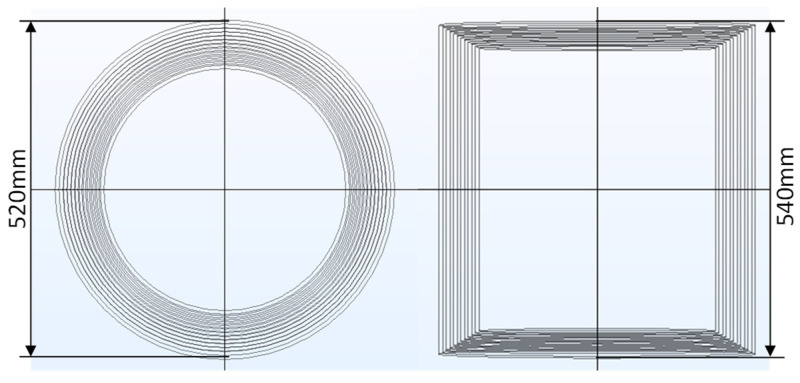
Two-dimensional structure of magnetic shielding barrel established by COMSOL.

**Figure 3 nanomaterials-14-00538-f003:**
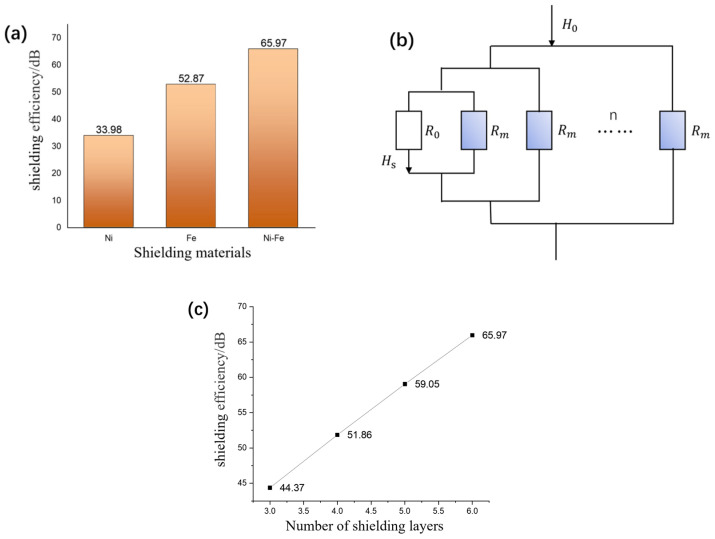
(**a**) Comparison of shielding efficiency of different shielding materials. (**b**) Comparison of shielding efficiency for different number of layers. (**c**) Schematic diagram of equivalent magnetic circuit of multi-layer shielding materials.

**Figure 4 nanomaterials-14-00538-f004:**
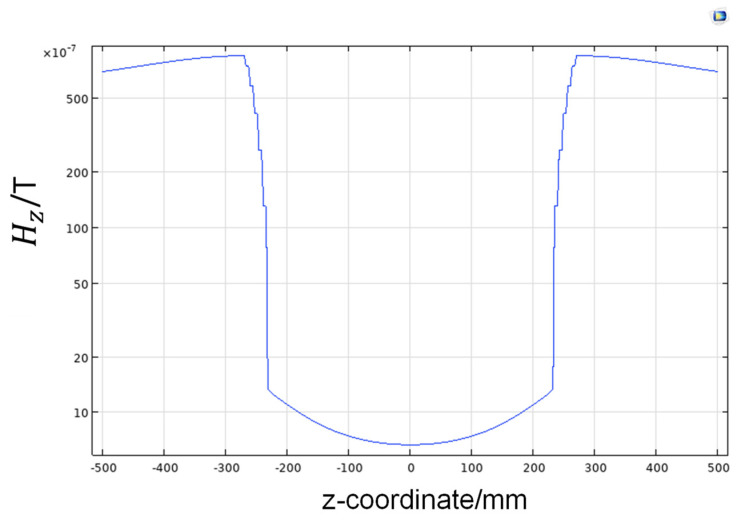
Results of shielding barrel compensation geomagnetic.

**Figure 5 nanomaterials-14-00538-f005:**
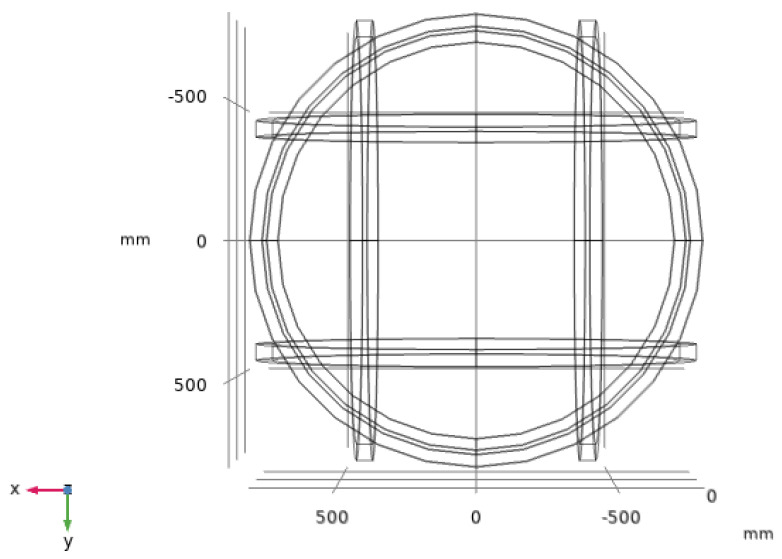
Structure of three-dimensional Helmholtz coils.

**Figure 6 nanomaterials-14-00538-f006:**
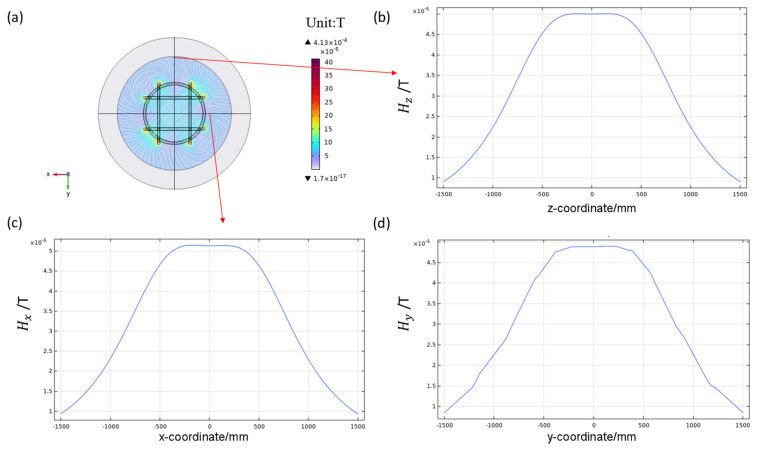
(**a**) Magnetic field distribution of three-dimensional Helmholtz coil. (**b**) Z-axis magnetic field Bz component. (**c**) X-axis magnetic field Bx component. (**d**) Y-axis magnetic field By component.

**Figure 7 nanomaterials-14-00538-f007:**
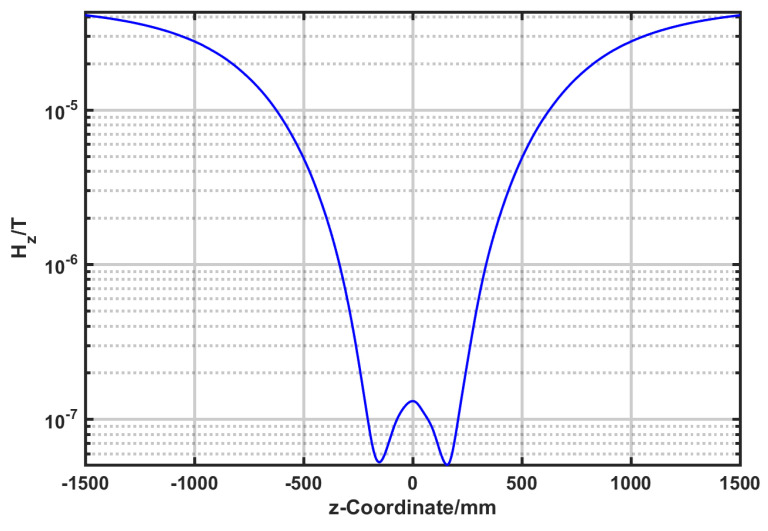
Helmholtz coil compensation geomagnetic results.

**Figure 8 nanomaterials-14-00538-f008:**
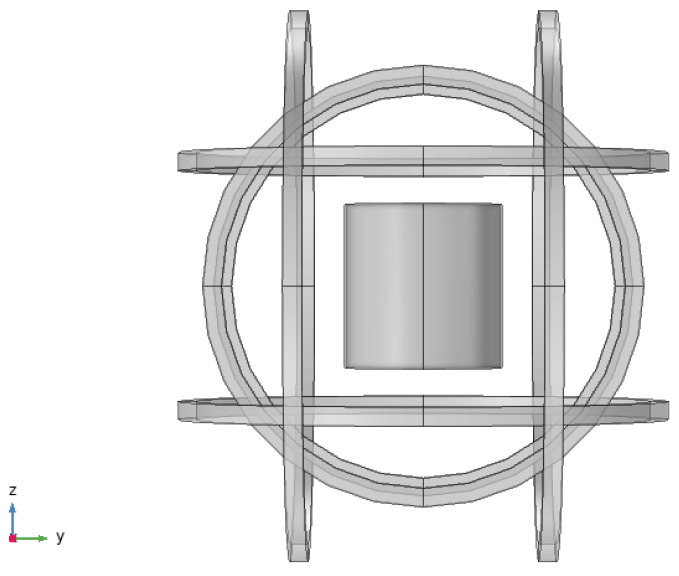
Schematic diagram of a magnetic shielding system combining active and passive modes.

**Figure 9 nanomaterials-14-00538-f009:**
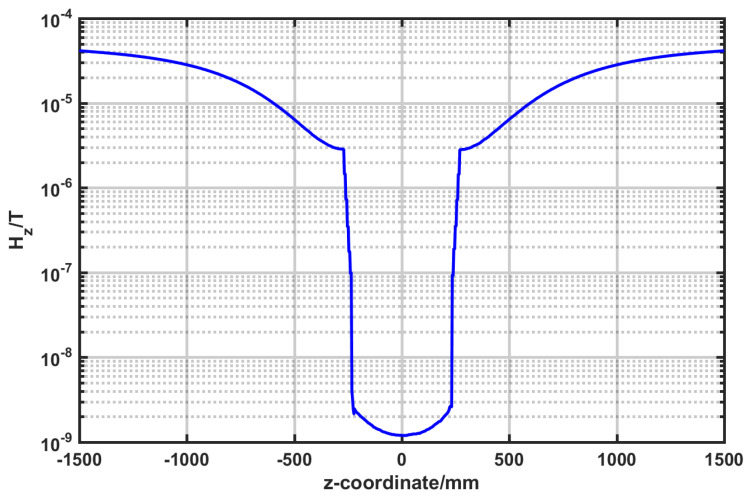
Magnetic field distribution for the geomagnetic magnetic field on the z-axis.

**Figure 10 nanomaterials-14-00538-f010:**
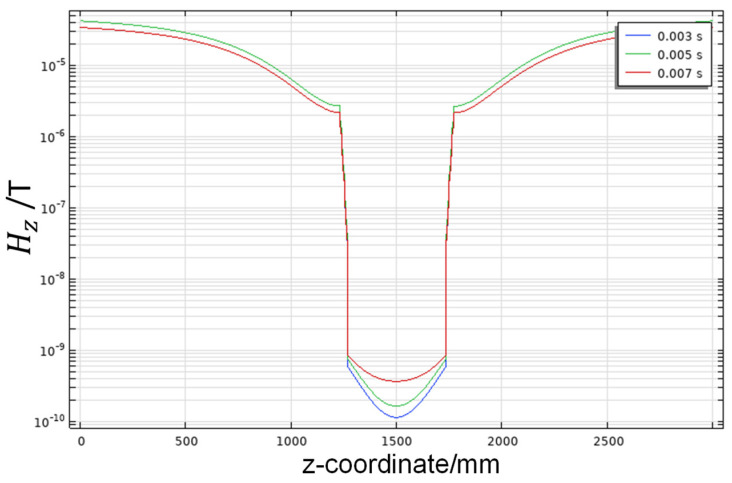
Transient analysis for the distribution of 50 Hz power-line magnetic field on the z-axis.

**Table 1 nanomaterials-14-00538-t001:** Comparison of magnetic shielding efficiency of different magnetic shielding systems.

Magnetic Shielding Material	Magnetic Shielding Method	Magnetic Shielding Efficiency	References
Single-layer polyimide material	Passive	26.1–28.8 dB	[17]
Ti_3_C_2_T_x_MXene/Ni thin film materials	Passive	41.7 dB	[18]
Single layer superconducting material	Passive	18–24.5 dB	[19]
Multilayer superconductivity/iron materials	Passive	53.5–70 dB	
Multi-layer permalloy	Active/Passive	113.98 dB	This work

## Data Availability

All data are contained within this article.

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
