# Peer review of "Theoretical Analysis of a Magnetic Shielding System Combining Active and Passive Modes"

_nanomaterials, 2024, doi:10.3390/nano14060538_

Round 1

Reviewer 1 Report

Comments and Suggestions for Authors

The article titled "Theoretical Analysis of a Magnetic Shielding System Combining Active and Passive Modes" presents a comprehensive study on an electromagnetic shielding system designed to mitigate interference from geomagnetic fields and 50Hz power-line frequencies. The authors propose a system integrating a three-dimensional Helmholtz coil with a multi-layer magnetic shielding barrel, using COMSOL simulation to evaluate its efficiency. They found that permalloy alloy outperforms pure iron and nickel in shielding effectiveness, and the system's overall efficiency improves linearly with the addition of multiple layers, achieving an impressive shielding efficiency of 113.98dB.

The article provides a robust theoretical foundation and simulation results, but it may benefit from experimental validation to confirm the simulation's predictions in real-world conditions.

While the article discusses the shielding efficiency of various materials, a deeper analysis of cost-effectiveness and practical implementation challenges of these materials in actual shielding systems would enhance its applicability.

In the manuscript must be add answer on these questions:

1. Have you considered or planned any experimental validation of the simulation results presented in this study?

2. How do the costs and practical implementation challenges of using permalloy alloy compare to other materials for electromagnetic shielding applications?

3. Can the proposed shielding system be scaled or adapted for use in other applications, such as medical devices or aerospace, where electromagnetic interference might be a concern?

4. Why autors describe only H parts of electromagnetic field and not electric part ? It is not needed ? For example authors in this paper investigated electric part around lines - https://ieeexplore.ieee.org/document/9269217 

5. The paper lacks a more extensive discussion on the results. There is a missing discussion on whether and how the simulation results would be beneficial in practice. What it would mean for practical applications. The cost of shielding and its potential usability are also not discussed.

6. The article provides a generally known statement - adding multiple layers of shielding material increases the SE value. With each additional increase in the number of shielding layers, the SE would increase. Does this addition of layers have limits?

7. The authors describe active shielding. How would it be feasible in practice? How would the power supply, consumption, etc., be implemented?

Reviewer 2 Report

Comments and Suggestions for Authors

Please add the answers to the following questions to the manuscript:

11-       While the simulation results demonstrate promising shielding efficiency, the practical implementation of such a system might encounter challenges such as cost, scalability, and integration with existing infrastructure. Can authors comment on the real-world applicability of their design?

22-   The study appears to focus on static conditions or a narrow range of frequencies (50Hz power-line frequency). However, real-world electromagnetic interference environments can be dynamic and encompass a broader frequency spectrum. How can authors explain evaluating the system's performance under dynamic conditions and a wider range of frequencies?

33-       The study does not address potential environmental impacts of the materials used in the shielding system. Factors such as material sourcing, manufacturing processes, and end-of-life disposal/recycling is crucial for ensuring the sustainability of the proposed solution must be addressed by the authors.

44-     The study does not discuss the robustness and reliability of the shielding system under various operating conditions and environmental factors such as temperature variations, humidity, and mechanical stresses. Authors need to assess on these elements.

55-       While the system demonstrates effectiveness in shielding interference magnetic fields, its impact on electromagnetic compatibility with nearby electronic devices and systems should be evaluated. Authors need to verify that the shielding system does not inadvertently disrupt the operation of surrounding equipment.

66-       Authors did not address potential safety hazards associated with the electromagnetic shielding system. Factors such as electromagnetic radiation exposure limits for human health, electrical safety, and fire hazards should be thoroughly evaluated to ensure the safety of personnel and equipment.

77-       Authors did not discuss the long-term performance degradation of the shielding system or the maintenance requirements to sustain its effectiveness over time. Factors such as material degradation, corrosion, and wear-and-tear must be considered for ensuring the longevity of the system.

Comments on the Quality of English Language

As above.

Reviewer 3 Report

Comments and Suggestions for Authors

Although the topic of the article may spark interest within the scientific community, this work presents significant shortcomings. 

1. The Introduction does not provide adequate background on which this article bases its scientific foundations; references should be implemented, especially regarding the state of the art in numerical analysis. 

2. Section 2 appears to be a mere repetition of notions already presented in previous papers and reviews (see Reference 11, which is not adequately cited in the text). 

3. The authors present this work as a study on the simulation analysis of the shielding capabilities of 3D structures, but the section dedicated to modelling (Section 3) does not provide any detail on the numerical model used, and the equations solved by the software; it merely lists the results obtained without any validation of the model. Furthermore, the results are reported in an approximate manner, without an efficient comparison between the results obtained with the different analysed shielding configurations. 

4. Just to mention, in Figure 6(a), units of measurement and the physical quantity shown are missing; the results reported in Figures 7 and 9 might suggest a poor quality of the mesh used for the 3D geometries, on which, however, the authors do not provide any information. 

5. Lastly, the conclusions are not supported by the results obtained, and I believe the authors do not provide enough motivations for the applicative validation of the proposed shielding solutions. I also note that after Section 3, the numbering is interrupted, jumping directly to the Conclusions section (Section 5). 

For all these reasons, I recommend rejecting this work.

Round 2

Reviewer 1 Report

Comments and Suggestions for Authors

Author Response

Thank you very much for the reviewer's time and efforts on our manuscript.

Reviewer 2 Report

Comments and Suggestions for Authors

Authors were expected to include their responses inside the manuscript as well. 

Comments on the Quality of English Language

As above

Reviewer 3 Report

Comments and Suggestions for Authors

The authors have implemented all the comments. However, in my opinion, the model could be presented in a more proper way; see for example 

-http://dx.doi.org/10.1063/1.4949516

-https://doi.org/10.1109/JSEN.2023.3313585
